# Use of Complementary and Alternative Therapies in People with Inflammatory Bowel Disease

**DOI:** 10.3390/ijerph21091140

**Published:** 2024-08-28

**Authors:** Laura Frank, Kelly Lambert

**Affiliations:** School of Medical, Indigenous and Health Sciences, University of Wollongong, Wollongong, NSW 2522, Australia

**Keywords:** complimentary therapies, cross sectional study, inflammatory bowel diseases, ulcerative colitis, Crohn’s disease, survey

## Abstract

Complementary and alternative medicines (CAMs) are frequently discussed by people with Inflammatory Bowel Disease (IBD). The aim of this study is to explore CAM use in Australians with IBD. This cross-sectional study was conducted via an anonymous online survey, predominantly distributed through IBD-specific social media accounts. Data collection occurred over a three-month period in 2021. Descriptive statistics, Chi-Square tests, and binary logistic regression were used to analyse quantitative data. A simple thematic analysis was conducted for qualitative free-text responses. Of the 123 responses, acupuncture (12.2%) and chiropractors (8.9%) were common CAM practitioners accessed. CAM practitioners were perceived to be ‘very helpful’ compared to mainstream health practitioners. The most common CAM products reported were vitamins (51.2%), probiotics (43.9%), and herbal medicine (30.9%). Common reasons for use were improved perceived improvements to wellbeing or for long-term management of IBD. Females were more likely to access CAM practitioners (OR 12.6, 95% CI 1.62–98.1, *p* = 0.02). Doctors were the participants’ primary source of information (64.2%), although many expressed dissatisfaction with conventional therapy and the desire for a more holistic approach to care. The use of CAMs in this sample was high. Limited research into the efficacy and safety of these therapies may prevent health professionals from discussing their use with patients. Improved communication with health professionals will allow patients to be active partners in their healthcare plans and can heighten patient satisfaction with conventional therapy.

## 1. Introduction

Inflammatory Bowel Disease (IBD) refers to chronic and relapsing inflammation of the gastrointestinal (GI) tract and includes the disorders Crohn’s Disease (CD) and Ulcerative Colitis (UC) [1]. Chronic inflammation impairs the GI tract’s functional ability, resulting in symptoms of persistent abdominal pain, nausea, vomiting, and diarrhoea [2,3]. IBD is uniquely characterised by distinct stages of exacerbation and remission, which often require repeat treatments and continuous monitoring [4,5]. As a result, individuals commonly experience high levels of psychological distress, fatigue, anxiety, and depression [2,6,7]. Many patients perceive there to be connections between such psychological factors and disease symptoms or relapse, although this relationship is not fully understood and is commonly overlooked in conventional therapy [2,6,7]. In addition, IBD can result in various extraintestinal manifestations, which commonly affect the skin, joints, eyes, and bones [6].

The medical management of IBD has experienced numerous advances in recent years, however, treatment options remain relatively limited [5,8]. The first line of treatment for IBD often involves potent pharmacological approaches such as antibiotics, steroids, immunosuppressants, and biologics [4,9]. While effective, these approaches can trigger various side effects, and as a result, medication nonadherence is common in IBD [10,11]. Where the disease is not able to be adequately managed by medication, surgical treatment may be offered to remove the affected areas of the bowel [12]. It has been estimated that surgical intervention will be required in up to 90% of individuals with CD and 30% of those with UC, with a significant proportion requiring repeat surgery [13,14]. While these treatments are proven to be effective in inducing remission, they place a high emotional and economic burden on those with IBD [15].

As with many chronic diseases, conventional therapy often falls short in managing the various indirect disease impacts. As such, Complementary and Alternative Medicines (CAMs) are increasingly used in addition to or in place of conventional medicine. CAMs are defined as “therapeutic practices which are not currently considered an integral part of conventional allopathic medical practice” [16]. The range of therapies included in this definition is extensive, and popular forms vary between countries [17]. Common types among those with IBD include dietary manipulation, nutritional supplements, herbalists, chiropractors, homeopaths, and acupuncture [17]. While there is minimal evidence to demonstrate efficacy, it has been estimated that up to half of those with IBD use some form of CAMs to help manage aspects of their disease [18].

The elusiveness of effective treatments for IBD may be a motivational factor for individuals pursuing CAMs. However, reasons for CAM use are generally unclear, and no qualitative research has been conducted within Australia to capture this perspective. A comprehensive survey conducted in 2011 among individuals with IBD found common reasons cited by survey respondents were in search for an optimum therapy, to terminate steroid medication, or to manage the side effects of conventional therapy [17]. Given the recent rise of CAM use in IBD, it is unclear if these suggested reasons for use hold true in the contemporary Australian population.

The progressive use of CAMs in IBD has also prompted substantial interest from healthcare professionals [19]. However, a lack of education in the area and limited evidence to support its use make many unprepared to counsel or discuss CAM use with their patients [17]. While healthcare professionals may recognise it as valuable information to gather, it has been reported that many do not routinely ask their patients about CAM use [19,20]. This, along with concerns over a negative response, has been consistently reported as reasons for individuals not revealing their CAM use to their healthcare providers [19,20,21]. As a result, nondisclosure of CAM use in IBD is high and often depends on the quality of the patient–doctor relationship [19,20,22]. Poor communication with health professionals has been found to be a predictor for the use of CAMs, as well as a common reason for individuals to seek advice from external, often unqualified sources [22]. Thus, open communication with patients is crucial in building and maintaining relationships and providing patient-centred care.

Various studies have attempted to capture the characteristics of CAM users in IBD, and identifying these may support health professionals to prompt discussions during routine care. Generally, a higher socioeconomic status, higher education, the female gender, and a younger age (under 40) have been associated with CAM use [23,24]. Furthermore, disease-associated predictors, including longer IBD duration, more severe disease activity, previous surgeries, and prolonged immunosuppressant or steroid use, may lead to CAM use [17,25,26,27]. However, there have been no recent studies focusing specifically on an Australian IBD cohort.

While evidence is growing to support the use of some CAMs in managing IBD symptoms, long-term implications and interactions with IBD medications remain largely unclear [2]. As a result, more robust research is needed to better understand the role of CAMs in IBD management, specifically in Australia. This will foster increased awareness of commonly used CAMs and allow for more education to be directed to professionals providing care for those with IBD. This is essential to enhance communication and, in turn, improve the overall quality of care. Thus, the aim of this study is to explore CAM usage in Australians with IBD, including common types, reasons for use, perceived efficacy in disease management, as well as the common characteristics of users and their sources of trusted information.

## 2. Materials and Methods

This study was approved by the joint Illawarra Shoalhaven Local Health District/University of Wollongong Human Research Ethics Committee (HREC number 2021/246). This study is reported in accordance with the Checklist for Reporting Results of Internet E-Surveys (CHERRIES) [28].

A cross-sectional study was conducted over a period of 10 months between January and November 2022. Inclusion criteria were any individual over the age of 18 with IBD (either Ulcerative Colitis or Crohn’s Disease). Individuals who were unable to complete the survey in English were excluded from the study.

The study was administered via an anonymous online survey. Informed and extended consent was obtained at the initiation of the survey, and tacit consent was assumed if completion of the survey occurred. There were no incentives offered, and respondents were provided with the option to exit the survey at any time should they no longer wish to participate. As per the Institutional Data Storage Policy, data obtained from this research was required to be stored on a password-protected drive (CloudStor) for five years before being securely destroyed.

The survey questions were derived from the International Complementary and Alternative Medicine Questionnaire (I-CAM-Q) [29]. The I-CAM-Q is a validated tool created to collect internationally comparable data on CAM usage. The tool contains four main domains, including types of health care providers, CAM therapies received from physicians, the use of herbal medicine and dietary supplements, and self-help practices.

Some amendments were made to the original questionnaire to make the survey more consistent with contemporary Australian CAM practice [30] and the Australian healthcare context. These amendments included additional responses to the question regarding reasons for the use of CAMs (‘to terminate or avoid medication’, ‘to manage treatment side effects’, and ‘symptom relief’). The option of a dietitian, nutritionist and reiki were also added as types of health practitioners. Vitamins, minerals, and probiotics were included as additional types of CAM products used, while manipulation, self-help practices, and spiritual healing were removed. We also added an additional question to our survey to ascertain the sources of information about IBD based on previous research by Wong et al., who found a strong desire for more information in people with IBD [30]. Free text boxes were added to each question to capture additional information from participants if they desired.

A total of 24 questions were included in the final survey with six domains covered, including individual and IBD characteristics, health literacy, types of CAM products and providers used, reasons for use, perceived efficacy, and sources of IBD information (Appendix A). Prior to distribution, the face validity of the survey was tested by the members of the research team with IBD (KL and LF).

Recruitment occurred via three main mechanisms:(1)Social media

Participants were recruited through snowball sampling via social media, including Instagram, Facebook, and Twitter. During recruiting, Australian IBD groups, pages, and accounts were primarily targeted to obtain local data. This method of recruitment was deemed most suitable as the onset of IBD commonly occurs between the ages of 15 and 35, and social media use in this age group is high [31].

(2)IBD organisations

Various IBD organisations shared the survey with members via their social media accounts or member newsletters. These are shown in Table 1.

(3)Professional networks

The survey was also distributed via local IBD specialist dietitians at two major teaching hospitals in New South Wales (St George and Wollongong Hospitals), as well as via professional networks of gastroenterologists participating in the longitudinal cohort Australian Inflammatory Bowel Disease Microbiome study [1].

A greater emphasis was placed on targeting Australians with IBD during recruitment. Due to lower-than-anticipated response rates, recruitment was extended to include a range of international IBD organisations and social media pages. Table 1 represents the individuals, groups, and organisations that distributed the survey.

For consistency, a standard message was used to contact organisations and social media administrators. Once consent from administrators was obtained, a standard announcement and image were distributed across Australian and relevant international social media channels.

Data collection occurred for a total of 84 days between 4 April and 27 June 2022. Qualtrics software (version July 2022) was used to create the survey and collate responses (Qualtrics, Provo, UT, USA). This survey platform facilitated the use of adaptive questioning, which was incorporated to reduce the complexity and duration of the survey. As a result, the number of items displayed per page ranged from 1 to 13 and varied based on the number of CAM products and providers selected. Respondents were unable to review responses prior to completing the survey or retract their responses after submission due to anonymity.

We determined that a survey sample size of 96 people was required using an estimated IBD population prevalence of 80,000, a 95% confidence interval, and a 10% margin of error (https://www.qualtrics.com/blog/calculating-sample-size/; Accessed on 27 August 2024). Responses in Qualtrics were exported into Excel. Data analysis was conducted using IBM SPSS statistics software (version 28.0.1.1 Armonk, NY, USA: IBM Corp). The Shapiro–Wilk test was conducted to assess the data for normality. Descriptive statistics, including median and range, were used to report the age of diagnosis. Categorical variables, including age, gender, disease type, types of CAM used, and information sources and needs, were reported as counts and percentages. The Chi-Square test was used to determine if there were differences in the proportions of responses according to disease type (CD vs. UC). Where cells had expected frequencies less than 5, Fisher’s Exact Test was used. Binary logistic regression was performed to ascertain predictors of CAM users such as age, gender, education level, surgery history, and country. In all analyses, a *p*-value of less than 0.05 was considered statistically significant.

Inductive thematic analysis was undertaken to explore free-text responses and identify common themes. All responses were reviewed individually to identify initial codes, and themes were generated through author discussions.

## 3. Results

A total of 135 survey responses were obtained during data collection. We are unable to estimate the response rate due to imprecise estimates of social media group membership, and some individuals may be members of more than one group. Five participants were excluded as they did not meet eligibility criteria (that is, have a diagnosis of IBD), along with seven incomplete responses. Thus, 123 responses were included in the data analysis.

### 3.1. Demographic Characteristics

Participant characteristics are presented in Table 2. The cohort consisted of 60 participants with Crohn’s Disease (CD) and 63 with Ulcerative Colitis (UC). The median age of diagnosis was 27 years (IQR: 18.5–35.5; range 9–74). Females were the most common gender reported among participants (82.9%), which did not differ between CD or UC (*p* = 0.12). A bachelor’s degree was the highest level of education reported by the majority of participants (35.8%). Half the participants responded from outside of Australia (52.9%), with the United Kingdom being the most frequently reported country of origin (21.2%). The proportion of participants with prior surgery was significantly higher in the CD group compared to UC (*p* < 0.001). Those with CD were more likely to be receiving biological therapy, while those with UC were more likely to be receiving steroids and 5ASA to manage their IBD (*p* = 0.02, *p* = 0.04, *p* < 0.001, respectively). The results from the three health literacy screening questions were added up and provided an average score. The average score of 1.3 indicates adequate health literacy for the cohort, with a score greater than or equal to 6 representing inadequate health literacy [32].

### 3.2. Use of CAM Practitioners

The most commonly visited mainstream health practitioners were medical specialists (Table 3, 80.5%) and general practitioners (59.3%). ‘Long-treatment and management of IBD’ was selected as the primary reason for the use of both specialists and dietitians (45.5% and 50.0%). Prescription renewals and blood tests were common reasons reported for visiting GPs (6.8% for both). Participants who utilised GPs, specialists, and dietitians perceived this to be ‘somewhat helpful’ (43.8%, 38.4% and 35.7%). Of the 10 participants who visited a nutritionist, 40% perceived the encounter to be ‘very helpful’.

Among the CAM therapists, ‘other’ was most frequently selected, followed by acupuncture (Table 3, 20.3% and 12.2% respectively). Free-text responses from those who selected ‘other’ indicated that naturopaths were most commonly visited (5%), followed by exercise physiologists, massage therapists, and osteopaths (1.7% for all). The primary reason for visiting a chiropractor, herbalist, or reiki practitioner was ‘to improve wellbeing’ (54.5%, 50.0%, and 75.0%, respectively). Participants perceived chiropractors, homeopaths, herbalists, and reiki practitioners to be ‘very helpful’ in managing their IBD (63.6%, 60.0%, 66.7% and 75.0%). Of those who utilised acupuncture, equal proportions perceived the therapy to be ‘very helpful’ or ‘somewhat helpful’ (46.7% each). No differences were found between the use of CAM practitioners, shown in Table 3.

### 3.3. Use of CAM Products

A total of 25 participants (20.3%) reported using no CAM products. From those who used CAM products, the most commonly used CAMs were vitamins (51.2%), probiotics (43.9%), and herbal medicine (30.9%). Of those using vitamins, vitamin D and B (either B6, B12, or complex) were the most commonly reported types (54.0% and 33.3%). The most frequently reported herbal medicine was turmeric (48.6%), followed by cannabis (27.0). Protein supplementation was commonly reported as an alternative nutritional supplement used by participants (72.4%).

The most frequently selected reasons for the use of vitamins were ‘to improve wellbeing’ (49.2%) and for ‘long-term treatment or management of IBD’ (27.0%). Improving wellbeing was the most frequently selected reason for using CAM products, including minerals (30.8%) and probiotics (29.6%). Long-term treatment also ranked highly as a reason for using herbal medicine (36.8%), probiotics (46.3%), minerals (30.8%), and other nutritional supplements (34.5%). No participants used herbal medicine, vitamins, minerals or homeopath remedies to manage an acute exacerbation of IBD (0.0% for all).

Participants perceived the use of most CAM products to be ‘very helpful’, with the majority selecting this for herbal medicine (44.7%), minerals (46.2%), nutrition supplements (51.7%), and other CAMs (54.5%). An equal proportion of participants who used vitamins believed these were very or somewhat helpful in managing IBD (42.9% for both). No differences were found between the use of CAM products according to disease type, as shown in Table 4.

### 3.4. Predictors for Use of CAM

Binary logistic regression was performed to determine predictors of CAM use. Age, gender, education, country of origin, previous surgery, and type of IBD were included in the model. Univariate analysis indicated that gender was the only significant predictor. A third gender/non-binary responses were excluded from the analysis. All other variables were not significant and were excluded from the final model.

The model for CAM practitioners and any CAM use correctly classified 64.5% and 84.0% of cases. Females were 12.6 times more likely to use a CAM practitioner (OR 12.6, 95% CI 1.62–98.1, *p* = 0.02) compared to males. The female gender was also associated with an increased likelihood of using any form of CAMs (OR 3.16, 95% CI 1.02–9.77, *p* = 0.046). Gender was not a predictor for the use of CAM products (OR 2.15, 95% CI 0.72–6.4, *p* = 0.17).

### 3.5. Information Sources and Needs

The primary source of IBD information was reported as doctors or medical specialists (64.2%) (Table 5). Media such as websites ranked second (27.6%), followed closely by IBD support groups (26.0%) and others with IBD (25.2%). There were no differences between the type of IBD and sources of information.

Overall, participants indicated they would like more information on nutrition and diet (45.5%), and this was more common in those with UC compared to CD (*p* = 0.01). The second most commonly selected information need was medication side effects (35.8%).

### 3.6. Thematic Analysis of Free Text Responses

Four main themes were evident from the analysis of free-text responses. Exemplar quotes were chosen to illustrate these key themes below.

#### 3.6.1. Theme 1: Reasons for Using CAM

Many participants reported using CAM when options for conventional medication were exhausted and their doctors were unable to provide alternative treatments. Participants also expressed using CAMs as an adjunct to conventional therapy to improve symptom management and overall quality of life.


*“While mesalamine has helped, having 3–4 tablespoons of Saurkraut first thing in the morning on an empty stomach significantly decreases and sometimes completely stops bleeding.”*



*“I think complimentary/alternative medicine has appealed to me more since I get frustrated knowing I’ll always have this condition. Sometimes I just want to try something different to feel better”*



*“It is hard to know which part is having the most effect. But the supplements and herbs seem to make much more of a difference to my IBD and overall health than the mesalazine does”*


#### 3.6.2. Theme 2: Perceived Efficacy of CAMs

Many participants perceived that there was a noticeable reduction in symptoms, which they attributed to various CAMs, including supplements, herbs, and stress management. Participants frequently suggested dietary changes were major contributors to symptom relief. The dietary changes reported include the exclusion of processed foods, gluten, dairy, and sugar, or including probiotic-rich foods, following an anti-inflammatory, vegetarian, or Crohn’s Disease Exclusion Diet (CDED).


*“Cryotherapy is expensive, so I can’t do it all the time, but find it to be the best at relieving pain and reducing effects of inflammation.”*



*“I changed to vegetarian diet and symptoms/hospitalisations have greatly reduced.”*



*“Along with my strict healthy diet (no processed foods) the supplements have really helped me half my conventional medicine dose and I feel much healthier and stronger.”*


#### 3.6.3. Theme 3: Dissatisfaction with Conventional Therapy

Many participants appeared frustrated about the conflicting attitudes on the use of CAMs among conventional practitioners. Participants often expressed feelings of helplessness that accompany living with a chronic disease. They also reported frustration with the perceived lack of treatment options, with one participant stating there are no alternatives if conventional treatment fails.


*“It’s only natural when you’re not having success with prescribed medication to turn to community and try to find complimentary options to keep yourself out of a situation that’s just so horrible and so reducing on your life”*



*“When your body fails you try everything and anything.”*


#### 3.6.4. Theme 4: A Desire for a Holistic Approach to IBD Care

Participants expressed a strong desire to incorporate both traditional and alternative care into IBD management. However, many stated they do not discuss CAMs with doctors and felt many doctors did not approve of CAM usage. Many perceived conventional practitioners to lack knowledge of the use of alternative therapies in IBD. Some participants perceived that doctors were unaware of the non-medical impact of IBD, such as the impact on mental health. There was a clear desire for more open dialogue to take place with conventional healthcare practitioners on the use of CAMs.


*“I find that traditional health professionals fail to treat people with IBD as a whole human. They are often less aware of the long list of issues we face… They are also close minded to holistic health which is sad as it often helps at least manage the extra intestinal manifestations of the disease. Holistic providers are willing to work with doctors and often help profoundly with mental health as they don’t stigmatise their patients like many doctors do and they really want to help with your quality of life.”*



*“It’s hard to find a doctor who is interested and knowledgeable about both biologics and supportive alternative health care.”*



*“Gastroenterologist was not helpful with questions about regarding probiotics and microbiomes”*


Other less prominent themes included:

#### 3.6.5. Theme 5: A Desire for More Research into the Efficacy of CAM


*“We also need more scientific studies done to show the effectiveness. So little money is put into IBD treatment compared to other disease.”*


#### 3.6.6. Theme 6: High Costs Preventing Individuals from Using CAM


*“I wish this was more widely accepted as a normal method of treatment and then there might be some medical costs coverage too. Costs can be a factor.”*


#### 3.6.7. Theme 7: Individuals Found No Benefits When Using CAM


*“Over the last 30 years I’ve tried many different alternative provision reiki healing. Reflexology crystal cleansing, hypnotherapy. I still got sick.”*


## 4. Discussion

The findings from this study suggest the widespread use of CAMs, with the majority (84%) of participants using various forms to manage aspects of their IBD. Among the most frequently reported types of CAMs were vitamin D, vitamin B (B6, B12 or complex), probiotics, herbal products, protein supplementation, and acupuncture. Perceptions of the efficacy of CAMs varied, and many expressed a desire for more conclusive research into its use in IBD. However, current users frequently reported CAMs to be ‘very helpful’ in managing certain disease manifestations. Aside from the female gender, no other characteristics were found to be predictors of CAM usage. However, the primary drivers for using CAMs were identified as long-term disease management and improved overall wellbeing.

This study extended upon prior research conducted by others into the information needs and sources for individuals with IBD [30,33,34]. A strong desire for nutrition and diet related information was identified, despite only a small proportion of participants using the services of a dietitian. The majority (64.2%) of respondents reported obtaining IBD information from their doctors or specialists. Participants shared a mutual dissatisfaction with conventional therapy, including IBD professionals for lacking knowledge of CAMs, as well as a willingness to openly discuss its potential role in disease management. As a result, the desire for a more holistic approach incorporating both conventional medicine and CAMs into overall IBD care was frequently reported.

The results of this study confirm earlier findings that probiotic use is increasing along with the market’s rapid growth and public interest in gut health [35]. The efficacy and safety of probiotics in IBD remains controversial, and currently, there is no evidence to support its use in CD [2,35,36,37,38]. On the contrary, multiple studies have found beneficial effects of some strains for inducing [35,36,37,38] and occasionally maintaining remission in active UC [2]. There has been minimal research into the effects of probiotics from a patient’s perspective. However, participants in this study mostly reported positive effects. While the research on probiotics may be promising, complications associated with their use have been found in the cases of severely immunocompromised patients, and interactions with commonly used IBD treatments, such as immunomodulatory drugs, have not been well researched [39,40].

The use of herbal products has been frequently reported as a common CAM therapy in IBD [2,17,24,41,42]. Many participants in this study reported supplementing with a compound found in turmeric known as curcumin [2]. These results are unsurprising given various studies suggest curcumin may be beneficial in inducing remission in active UC [2,43]. Interestingly, cannabis and its derivatives were reported as commonly used herbal therapies in this, and recent studies [18,44]. This could be attributed to its progressive legalisation, along with increased awareness and utilisation of its medicinal properties. A recent Australian study suggested one-quarter of individuals were currently or had previously used cannabis to manage their IBD [44]. Individuals using cannabis for IBD have reported improved sleep, reduced stress and relief in abdominal pain, nausea, and diarrhoea [44,45,46]. However, there is currently no evidence to support the safe use of cannabis in IBD, and the long-term effects on inflammation and disease progression are unclear. [2,18,46,47].

Dietary manipulation is common in IBD and often results in various food aversions and suboptimal dietary intake [48,49]. This places individuals at high risk of developing nutritional deficiencies and malnutrition [50]. Similar to prior research, participants in this study reported frequently avoiding dairy products, and calcium supplementation was uncommon [51]. Suboptimal calcium intake can contribute to low bone density in those with IBD, which predisposes individuals to developing osteopenia and osteoporosis [52]. Some participants also reported following a gluten-free diet, which typically contains comparatively less dietary fibre [53] and has not proven to have beneficial effects on disease progression [2,54]. Considering that current research indicates that suboptimal dietary fibre is common in those with IBD [50], particular attention should be placed on the diet quality of those following a gluten-free diet. These forms of dietary manipulation may be in part due to a desire to manage symptoms, as well as a lack of access to dietetic advice, with only a small proportion of participants in this study reporting working with a dietitian. Given that dietitians are the only professionals able to receive Medicare reimbursement for medical nutrition therapy in Australia, incorporating dietitians into care and encouraging stronger multidisciplinary relationships is essential to mitigate malnutrition risk and provide evidence-based advice to individuals with IBD.

A theme that remained consistent with past findings is dissatisfaction with conventional therapy, particularly the perception of a lack of alternatives if conventional medicine fails [17,42]. Until recently, many users reported exclusively relying on CAMs to manage their IBD and were often seeking a therapy that would allow termination or avoidance of conventional medication [17,41]. This may have contributed to some concerns raised by health professionals about CAM use, as reduced adherence to conventional medication is associated with worse disease outcomes [11]. However, this study confirms recent findings that individuals would now prefer to use CAMs in combination with conventional therapy rather than exclusively to treat or manage their IBD [20]. Similarly, a prior study found health professionals demonstrated positive attitudes about the use of CAMs when in conjunction with conventional therapy [19]. However, the study also found health professionals were interested in further education, as they felt unprepared to discuss CAMs if patients raised the topic, and as a result, many did not routinely ask individuals with IBD about CAM use [19,55]. The decision for individuals with IBD to use CAMs, as well as inform their health professionals, has been found to depend on the quality of the patient–doctor relationship [43]. Prior research has found that up to 72% of CAM users do not disclose their CAM use to their IBD health professionals [20].

Similar to previous studies, participants of this study reported feeling that their health professionals were reluctant to discuss CAMs, and they feared judgment if they raised the subject [20]. For this reason, many participants reported not discussing CAMs with their health professionals, although they expressed a desire for more communication about this topic to take place. A lack of communication of CAMs to health professionals can prevent individuals from making informed and shared decisions about their care, which could lead to adverse health outcomes. This can, in turn, compromise the quality of the patient–doctor relationship and, therefore, the delivery of patient-centred care.

There are various strengths to this study, including a high response rate. The survey used was adapted from prior research and a previously validated tool, which allowed results to be reflective of contemporary healthcare and modern CAM use in Australia. Furthermore, the face validity of the survey was tested by members of the research team with IBD. Additionally, recruiting IBD support groups and organisations allowed those most likely to be engaged in managing their IBD to participate in the study. The use of social media as a recruitment strategy may also be a limitation and exclude those with limited digital literacy. Expanding recruitment to include international responses created a limitation, as we did not achieve the required sample size to be generalisable to an Australian population. As the survey was limited to English speakers and participants were predominantly from English-speaking countries, the diversity of CAM use among cultures was also not explored, and cultural bias may be present. This is an important point, as variation in CAM use between ethnicities is known to occur [56]. Additionally, as most of the respondents identified as female, the extent to which gender influences CAM use would require further investigation. IBD is known to be more common in women due to sex-determined epigenetic factors related to the X chromosome [57]. Another limitation was the primary collection of quantitative data, which may not provide the insight that could be attained through predominantly qualitative methods. In response to the desire for more communication about CAM use expressed among those with IBD, the opportunity exists for health professionals to ask about CAM use during routine care. This may facilitate a dialogue between health professionals and patients about CAMs and lead to improved patient satisfaction with conventional therapy. Ongoing professional development focusing on the current evidence for commonly used CAMs and current guidelines for CAM use in IBD would allow health professionals to provide evidence-based advice that is aligned with patient needs. More collaborative health care may also create opportunities for future research that is patient-driven. Future qualitative research would be beneficial in expanding the current understanding of CAM use in IBD, particularly in an Australian setting. Furthermore, research should be directed towards the safety and efficacy of commonly used CAMs, especially when used in conjunction with IBD medication.

## 5. Conclusions

The widespread use of complementary and alternative therapies suggests clinicians must address usage as part of routine care. While many patients perceive CAMs to be beneficial in managing various disease symptoms, limited research into the efficacy and safety of CAMs in IBD exists. This may constitute a barrier for health professionals to discuss CAM with their patients. However, poor communication appears to be a major factor contributing to patients’ dissatisfaction with conventional therapy. It is, therefore, important for health professionals to be considerate of and knowledgeable in CAM use in IBD, especially given they are the primary source of trusted information. Stronger partnerships forged between patients, specialists, doctors, and dietitians may allow for the provision of treatment and management care plans that are appropriate to patients’ needs and preferences.

## Figures and Tables

**Table 1 ijerph-21-01140-t001:** Individuals, groups, and organisations that shared the survey and their channel of distribution.

Name (Instagram Handle)	Channel of Distribution	Number of Followers
Crohn’s & Colitis Australia	LinkedIn Facebook	826 22,565
IBD & IBS Patient Advocate (@betterbelliesbymolly)	Instagram	4734
Gut Health & IBS Dietitians (@fodmapdietetics)	Instagram	2918
IBD Relief (@ibdrelief)	Instagram Internal newsletter	2052 Unknown
Charlotte Kate (@thecrohnsbaker)	Instagram	290
The Gut-Friendly Dietitian (@thegutfriendlydietitian)	Instagram	2497
Crohn’s Colitis New Zealand Canterbury Support Group	Facebook	397
Crohn’s & Colitis UK	Twitter	41,500
Rachel|Crohn’s & Colitis (@beyond.IBD)	Instagram	2558

**Table 2 ijerph-21-01140-t002:** Demographic characteristics of survey participants.

	Type of IBD			*p*-Value
	CD n = 60 n (%)	UC n = 63 n (%)	Total n = 123 n (%)	
Age of diagnosis (years)				<0.001 *
Median	25	28	27	
Range	9–74	15–69	9–74	
Age (years)				0.18
18–24	6 (4.9)	2 (1.6)	8 (6.5)	
25–34	18 (14.6)	24 (19.5)	42 (34.1)	
35–44	14 (11.4)	15 (12.2)	29 (23.6)	
45–54	6 (4.9)	12 (9.8)	18 (14.6)	
>55	16 (13.0)	10 (8.1)	26 (21.1)	
Gender				0.12
Male	12 (9.8)	7 (5.7)	19 (15.4)	
Female	46 (37.4)	56 (45.5)	102 (82.9)	
Third gender	2 (1.6)	0 (0.0)	2 (1.6)	
Level of education				0.26
High school	10 (8.1)	8 (6.5)	18 (14.6)	
Trade qualification	10 (8.1)	8 (6.5)	18 (14.6)	
Bachelor degree	22 (17.9)	22 (17.9)	44 (35.8)	
Master’s degree	12 (9.8)	10 (8.1)	22 (17.9)	
PhD	0 (0.0)	5 (4.1)	5 (4.1)	
Other	6 (5.0)	10 (8.1)	16 (13.0)	
Country				0.06
Australia	33 (27.3)	24 (19.8)	57 (47.1)	
Outside of Australia	26 (21.5)	38 (31.4)	64 (52.9)	
Previous surgery				<0.001 *
Yes	31 (25.2)	2 (1.6)	33 (26.8)	
No	29 (23.6)	61 (49.6)	90 (73.1)	
Medications				
Antibiotics	2 (1.7)	3 (2.5)	5 (4.1)	0.52
Steroids	6 (5.0)	15 (12.2)	21 (17.1)	0.04 *
5 ASA	8 (6.5)	39 (31.7)	47 (38.2)	<0.001 *
Immunosuppressants	14 (11.4)	13 (10.6)	27 (22.0)	0.72
Biological therapy	28 (22.8)	17 (13.8)	45 (36.6)	0.02 *
Other	15 (12.2)	17 (13.8)	32 (26.0)	0.80
Health literacy score				
Mean	1.3	1.2	1.3	
SD	0.16	0.14	0.15	
How confident are you in filling out forms by yourself?				0.09
Extremely—1	53 (43.1)	60 (48.8)	113 (91.9)	
Quite a bit—2	7 (5.7)	2 (1.6)	9 (7.3)	
Somewhat—3	0 (0.0)	1 (0.8)	1 (0.8)	
A little bit—4	0 (0.0)	0 (0.0)	0 (0.0)	
Not at all—5	0 (0.0)	0 (0.0)	0 (0.0)	
How often do you have someone help with hospital materials?				0.69
Never—1	45 (36.6)	51 (41.5)	96 (78.0)	
Occasionally—2	10 (8.1)	9 (7.3)	19 (15.4)	
Sometimes—3	3 (2.4)	1 (0.8)	4 (3.3)	
Often—4	0 (0.0)	1 (0.8)	1 (0.8)	
Always—5	2 (1.6)	1 (0.8)	3 (2.4)	
How often do you have problems learning about your medical condition?				0.78
Never—1	44 (36.4)	50 (40.7)	94 (76.4)	
Occasionally—2	11 (8.9)	9 (7.3)	20 (16.3)	
Sometimes—3	4 (3.3)	2 (1.6)	6 (4.9)	
Often—4	0 (0.0)	1 (0.8)	1 (0.8)	
Always—5	1 (0.8)	1 (0.8)	2 (1.6)	

Data is represented as number and % of total participants n = 123; Data is divided into subgroups CD (Crohn’s Disease), UC (Ulcerative Colitis) and Total (Both Crohn’s Disease and Ulcerative Colitis); Health literacy score of ≥6 indicates inadequate health literacy. 5ASA: 5-aminosalicylic acid or mesalazine. * represents *p* < 0.05

**Table 3 ijerph-21-01140-t003:** The use of complementary and alternative health practitioners among participants.

Types of Practitioners										
	Mainstream Health Practitioners n (%)				Complementary and Alternative Therapists n (%)					
	GP	Specialist	Dietitian	Nutritionist	Chiropractor	Homeopath	Herbalist	Reiki	Acupuncture	Other
Visited in previous 12 months	73 (59.3)	99 (80.5)	28 (22.8)	10 (8.1)	11 (8.9)	5 (4.1)	6 (4.9)	4 (3.3)	15 (12.2)	25 (20.3)
**Reasons for use**										
Acute exacerbation	7 (9.6)	15 (15.2)	0 (0.0)	1 (10.0)	1 (9.1)	1 (20.0)	0 (0.0)	0 (0.0)	2 (13.3)	0 (0.0)
Long-term treatment/management	10 (13.7)	45 (45.5)	14 (50.0)	2 (20.0)	1 (9.1)	1 (20.0)	0 (0.0)	1 (25.0)	1 (6.7)	5 (20.0)
Overall wellbeing	10 (13.7)	0 (0.0)	9 (32.1)	5 (50.0)	6 (54.5)	0 (0.0)	3 (50.0)	3 (75.0)	4 (26.7)	7 (28.0)
Terminate/avoid medication	1 (1.4)	1 (1.0)	0 (0.0)	1 (10.0)	0 (0.0)	1 (20.0)	1 (16.7)	0 (0.0)	1 (6.7)	0 (0.0)
Manage treatment side effects	6 (8.2)	1 (1.0)	0 (0.0)	0 (0.0)	0 (0.0)	0 (0.0)	0 (0.0)	0 (0.0)	1 (6.7)	2 (8.0)
Symptom relief	8 (11.0)	7 (7.1)	1 (3.6)	0 (0.0)	2 (18.2)	1 (20.0)	1 (16.7)	0 (0.0)	4 (26.7)	5 (20.0)
Other	24 (32.9)	18 (18.2)	2 (7.1)	0 (0.0)	0 (0.0)	0 (0.0)	0 (0.0)	0 (0.0)	1 (6.7)	4 (16.0)
How helpful was it										
Very helpful	20 (27.4)	35 (35.4)	8 (28.6)	4 (40.0)	7 (63.6)	3 (60.0)	4 (66.7)	3 (75.0)	7 (46.7)	8 (32.0)
Somewhat helpful	32 (43.8)	38 (38.4)	10 (35.7)	3 (30.0)	2 (18.2)	1 (20.0)	1 (16.7)	1 (25.0)	7 (46.7)	11 (44.0)
Not at all helpful	7 (9.6)	8 (8.1)	8 (28.6)	2 (20.0)	1 (9.1)	0 (0.0)	0 (0.0)	0 (0.0)	0 (0.0)	2 (8.0)
Unsure	2 (2.7)	3 (3.0)	1 (3.6)	0 (0.0)	0 (0.0)	0 (0.0)	0 (0.0)	0 (0.0)	0 (0.0)	1 (4.0)
Other	4 (5.5)	3 (3.0)	0 (0.0)	0 (0.0)	0 (0.0)	0 (0.0)	0 (0.0)	0 (0.0)	0 (0.0)	0 (0.0)
How often in past 3 months										
Weekly	3 (2.4)	0 (0.0)	0 (0.0)	1 (10.0)	0 (0.0)	0 (0.0)	1 (16.7)	0 (0.0)	3 (20.0)	3 (12.0)
Fortnightly	2 (2.7)	2 (2.0)	0 (0.0)	0 (0.0)	2 (18.2)	0 (0.0)	0 (0.0)	0 (0.0)	1 (6.7)	0 (0.0)
Monthly	30 (41.1)	17 (17.2)	5 (17.9)	3 (30.0)	3 (27.3)	4 (80.0)	2 (33.3)	3 (75.0)	5 (33.3)	9 (36.0)
Other	31 (42.5)	68 (68.7)	21 (75.0)	5 (50.0)	5 (45.5)	0 (0.0)	2 (33.3)	1 (25.0)	5 (33.3)	11 (44.0)

Data is represented as number and % of total participants n = 123 for visited in previous 12 months and % of subgroup for reasons for use, helpfulness and frequency. A total of 79 (64.2%) of participants reported no use of CAM therapists.

**Table 4 ijerph-21-01140-t004:** The use of complementary and alternative products among participants.

Types of Complementary and Alternative Products							
	Herbal Medicine	Vitamins	Minerals	Probiotics	Other Nutritional Supplements	Homeopath Remedies	Other CAM
Most common types	Turmeric 18 (48.6) Cannabis 10 (27.0) Oregano oil 3 (8.1)	Vit D. 34 (54.0) Vit B. 21 (33.3) Multi. 18 (28.6)	Mg. 17 (68.0) Zinc. 13 (52.0) Iron. 10 (40.0)	-	Protein 21 (72.4) Omega-3 10 (34.5) Collagen 5 (17.2)	-	Diet. 4 (18.2) Exercise 3 (2.5)
Used in previous 12 months	38 (30.9)	63 (51.2)	26 (21.1)	54 (43.9)	29 (23.6)	5 (4.1)	22 (17.9)
Currently uses	24 (19.5)	58 (47.2)	20 (16.3)	44 (35.8)	19 (15.4)	3 (2.4)	10 (8.1)
Reasons for use							
Acute exacerbation	0 (0.0)	0 (0.0)	0 (0.0)	1 (1.9)	1 (3.4)	0 (0.0)	0 (0.0)
Long-term treatment/management	14 (36.8)	17 (27.0)	8 (30.8)	25 (46.3)	9 (31.0)	1 (20.0)	5 (22.7)
Overall wellbeing	8 (21.1)	31 (49.2)	8 (30.8)	16 (29.6)	10 (34.5)	1 (20.0)	4 (18.2)
Terminate/avoid medication	2 (5.3)	0 (0.0)	1 (3.8)	4 (7.4)	1 (3.4)	2 (40.0)	1 (4.5)
Manage treatment side effects	1 (2.6)	2 (3.2)	2 (7.7)	1 (1.9)	1 (3.4)	0 (0.0)	1 (4.5)
Symptom relief	9 (23.7)	6 (9.5)	5 (19.2)	5 (9.3)	3 (10.3)	0 (0.0)	2 (9.1)
Other	2 (5.3)	5 (7.9)	2 (7.7)	1 (1.9)	0 (0.0)	0 (0.0)	1 (4.5)
How helpful was it							
Very helpful	17 (44.7)	27 (42.9)	12 (46.2)	21 (38.9)	15 (51.7)	2 (40.0)	12 (54.5)
Somewhat helpful	13 (34.2)	27 (42.9)	10 (38.5)	23 (42.6)	8 (27.6)	2 (40.0)	1 (4.5)
Not at all helpful	1 (2.6)	1 (1.6)	0 (0.0)	2 (3.7)	1 (3.4)	0 (0.0)	0 (0.0)
Unsure	6 (15.8)	3 (4.8)	4 (15.4)	7 (13.0)	1 (3.4)	0 (0.0)	1 (4.5)
Other	0 (0.0)	1 (1.6)	0 (0.0)	0 (0.0)	0 (0.0)	0 (0.0)	0 (0.0)

Data is represented as number and % of total participants n = 123 for visited in previous 12 months and currently uses, and % of subgroup for reasons for use, helpfulness and frequency; a total of 25 (20.3%) of participants reported no use of CAMs; Turmeric includes the use of Curcumin; Cannabis includes the use of Cannabidiol (CBD oil) and Tetrahydrocannabinol (THC); Vitamin B includes the use of Vitamin 6, Vitamin 12 and Vitamin B complex; Omega 3 includes the use of Fish oil.

**Table 5 ijerph-21-01140-t005:** Source and type of information desired.

	Type of IBD		
	CD n (%)	UC n (%)	Total n (%)
Sources of information on IBD treatment/management			
Doctor/medical specialist	43 (35.0)	36 (29.3)	79 (64.2)
News article	9 (7.3)	10 (8.1)	19 (15.4)
Brochures	2 (1.6)	1 (0.8)	3 (2.4)
Health stores/health food shops	2 (1.6)	3 (2.4)	5 (4.1)
Pharmacy	2 (1.6)	2 (1.6)	4 (3.3)
Websites	21 (17.1)	13 (10.6)	34 (27.6)
Social media	12 (9.8)	11 (8.9)	23 (18.7)
Online blogs	3 (2.4)	1 (0.8)	4 (3.3)
Others with IBD	16 (13.0)	15 (12.2)	31 (25.2)
IBD Support groups	16 (13.0)	16 (13.0)	32 (26.0)
Journal articles	8 (6.5)	13 (10.6)	21 (17.1)
Other	4 (3.3)	2 (1.6)	6 (4.9)
Information needs			
Common IBD symptoms	5 (4.1)	8 (6.5)	13 (10.6)
Disease complications	13 (10.6)	9 (7.3)	22 (17.9)
Causes of IBD	17 (13.8)	14 (11.4)	31 (25.2)
Prognosis	15 (12.2)	13 (10.6)	28 (22.8)
Risk of developing cancer	17 (13.8)	15 (12.2)	32 (26.0)
Medication effects	24 (19.5)	20 (16.3)	44 (35.8)
Nutrition and diet	20 (16.3)	36 (29.3) *	56 (45.5)
Managing time away from school/work	10 (8.1)	11 (8.9)	21 (17/1)
Support sources	12 (9.8)	14 (11.4)	26 (21.1)
Connecting with others with IBD	9 (7.3)	10 (8.1)	19 (15.4)
Other	0 (0.0)	5 (4.1) *	5 (4.1)

Data reported as number and % of total participants n = 123; * indicates a significant *p*-value of <0.05.

## Data Availability

The data presented in this study are available on request from the corresponding author due to ethical reasons (ethics approval did not approve the submission of data to public data repositories).

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
