# Peer review of "Use of Complementary and Alternative Therapies in People with Inflammatory Bowel Disease"

_ijerph, 2024, doi:10.3390/ijerph21091140_

Round 1
Reviewer 1 Report
Comments and Suggestions for Authors
The authors study the use of Complementary and Alternative Therapies (CAM) for IBD patients. The study is well-presented, the authors discussed previous research studies. The motivation is sufficiently introduced. The result section contains the necessary numerical results. The discussion section is well written. However, I have some remarks for the authors:
1. Materials and Methods: the use of tactic consent needs to be explained. The paragraph presenting some amendments to the original questionnaire needs improvement to be clearly stated. I did not understand the sentence "Categorical variables including age, gender, disease type, types
of CAM used and information sources and needs were reported as counts and percentages". Same for the sentence "Where cells had expected frequencies less than 5, Fishers Exact Test was used."
3. Results: The female participant's number is 3 times more than the male's number. How do the authors think it impacted your results ? I think the authors need to discuss this element of consideration within their statistical analysis. Moreover, the age range was not clearly stated in the result analysis. Can the authors explain this statement "Those that utilised acupuncture, equally perceived the therapy to be ‘very helpful’ and ‘somewhat helpful’ (46.7% for both). No differences were found between use of CAM practitioners according to disease type and are shown in Table 2." Indeed, I have difficulty finding these results in Table 2. The authors found that gender was the only significant predictor. What technique did you use to find this result ? I recommend the authors to clarify this point. Again, as the female participant's number is 3 times more than the male participant's number, did the authors think it impacted their results/conclusions ?
Finally, the conclusion section should be improved.
Comments on the Quality of English Language
I have some comments:
1. Abstract: Can be improved, especially when presenting numerical results for ease of understanding for the reader.
2. Materials and Methods: when citing research by Wong et al., the citation number is missed. A numbering mistake exists for the third considered mechanism, i.e., Professional networks.
3. Results: Table 1, there is a formatting error within the raw "How confident are you filling out forms by yourself?"
4. Discussion: also a formatting error with citation (20).
Author Response
Materials and Methods: the use of tactic consent needs to be explained.
This is a spelling error and refers to tacit consent
The paragraph presenting some amendments to the original questionnaire needs improvement to be clearly stated. I did not understand the sentence "Categorical variables including age, gender, disease type, types of CAM used and information sources and needs were reported as counts and percentages". Same for the sentence "Where cells had expected frequencies less than 5, Fishers Exact Test was used."
These sentences refer to how data is presented and are reported in a manner consistent with statistical reporting in the scientific literature. No additional changes are made.
- Results:The female participant's number is 3 times more than the male's number. How do the authors think it impacted your results ? I think the authors need to discuss this element of consideration within their statistical analysis.
Women are known to be more likely to have IBD due to a genetic disposition related to the X chromosome. This means the survey findings of having more female than males is consistent with the epidemiology of IBD. This is now included in the discussion and reads: Additionally, as most of the respondents identified as female, the extent to which gender influences CAM use would require further investigation. IBD is known to be more common in women due to sex determined epigenetic factors related to the X chromosome[57].
Moreover, the age range was not clearly stated in the result analysis.
Age range is included in the results. It reads: The median age of diagnosis was 27 years (IQR: 18.5- 35.5; range 9-74).
Can the authors explain this statement "Those that utilised acupuncture, equally perceived the therapy to be ‘very helpful’ and ‘somewhat helpful’ (46.7% for both). No differences were found between use of CAM practitioners according to disease type and are shown in Table 2." Indeed, I have difficulty finding these results in Table 2.
We have revised wording to state: Those that utilised acupuncture, equal proportions perceived the therapy to be ‘very helpful’ or ‘somewhat helpful’ (46.7% each).
The authors found that gender was the only significant predictor. What technique did you use to find this result ? I recommend the authors to clarify this point. Again, as the female participant's number is 3 times more than the male participant's number, did the authors think it impacted their results/conclusions ?
Binary logistic regression was used as outlined in the methods section (Binary logistic regression was performed to ascertain predictors of CAM users such as age, gender, education level, surgery history and country.). After controlling for this in the regression model, female gender still predicted likelihood of using a CAM Practioner but not CAM product.
Finally, the conclusion section should be improved.
This has been amended.
- Materials and Methods: when citing research by Wong et al., the citation number is missed. A numbering mistake exists for the third considered mechanism, i.e., Professional networks.
This has now been amended thank you
- Results: Table 1, there is a formatting error within the raw "How confident are you filling out forms by yourself?"
Now amended
- Discussion: also a formatting error with citation (20).
Now amended
Reviewer 2 Report
Comments and Suggestions for Authors
This manuscript analyses the use of CAM in the Australian IBD population by conducting a multidimensional questionnaire survey of Australian residents. This is a relatively complete qualitative and quantitative study, which is meaningful for understanding the application of complementary and alternative medicine in Australia.
This study has a serious failure in design. The ethnicity of the study subjects was not reported in the questionnaire results. I don't know if this is in the questionnaire, if so, it should be analyzed, if not, it should be discussed.
Author Response
This study has a serious failure in design. The ethnicity of the study subjects was not reported in the questionnaire results. I don't know if this is in the questionnaire, if so, it should be analyzed, if not, it should be discussed.
Ethnicity was not captured in the survey as it was not modified from the original survey by Wong et al. This is a useful point as CAM use may be associated with particular ethnicities. This has been included in the limitations. It now reads : As the survey was limited to English speakers and participants were predominantly from English speaking countries, the diversity of CAM use among cultures was also not explored and cultural bias may be present. This is an important point as variation in CAM use between ethnicities is known to occur [56].
Reviewer 3 Report
Comments and Suggestions for Authors
This manuscript is very informative as the diseases of GI track and especially inflammatory bowel disease are in rise. Updated information on the CAM therapies is very useful. Health professional should be aware of the alternative ways of treating the symptoms and this paper would provide insight to it.
The manuscript is well-written.
Comments/suggestions:
1. There are two tables as Table 1.
2. What is 5 ASA - could it be spelled out or explained in footnote of the table?
3. Difference between dietitian and nutritionist, is the training similar in Australia vs. in other participating countries?
4. Some discussion on the Social Media as a mean of reaching CAM users would be needed (limitations). Are all age groups equally represented? Should there be also 55-64, and > 65 instead of combining all over 55 years into the same group?
5. I agree with authors that applying more qualitative research would have provided additional insight into the topic like perceptions of CAM by age (just a comment).
I would have expected some discussion about the possible limitations of using
Author Response
- There are two tables as Table 1.
Errors with labelling tables have been amended
- What is 5 ASA - could it be spelled out or explained in footnote of the table?
This has been included
- Difference between dietitian and nutritionist, is the training similar in Australia vs. in other participating countries?
In Australia, only dietitians can receive Medicare reimbursement for patient consultations. It now reads: Given that dietitians are the only profession able to receive Medicare reimbursement for medical nutrition therapy in Australia, incorporating dietitians into care and encouraging stronger multidisciplinary relationships is essential to mitigate malnutrition risk and provide evidence based advice to individuals with IBD.
- Some discussion on the Social Media as a mean of reaching CAM users would be needed (limitations).
This is an excellent point. We have now included the following: The use of social media as a recruitment strategy may also be a limitation and exclude those with limited digital literacy.
- Are all age groups equally represented? Should there be also 55-64, and > 65 instead of combining all over 55 years into the same group?
We are unable to report finer details on age as the data collection in the survey included 55 and over only
I would have expected some discussion about the possible limitations of using
No change made as comment unclear